# Low educational level increases functional disability risk subsequent to heart failure in Japan: On behalf of the Iwate KENCO study group

Shuko Takahashi[1,2,3]*, Kozo Tanno[4], Yuki Yonekura[5], Masaki Ohsawa[6], Toru Kuribayashi[7], Yasuhiro Ishibashi[8], Shinichi Omama[9], Fumitaka Tanaka[10], Toshiyuki Onoda[11], Kiyomi Sakata[4], Makoto Koshiyama[12], Kazuyoshi Itai[13], Akira Okayama[14]

1 Division of Medical Education, Iwate Medical University, Shiwa-gun, Iwate, Japan, 2 Takemi Program in International Health, Harvard T.H. Chan School of Public Health, Boston, MA, United States of America, 3 Department of Health and Welfare, Iwate Prefectural Government, Morioka, Iwate, Japan, 4 Department of Hygiene and Preventive Medicine, School of Medicine, Iwate Medical University, Shiwa-gun, Iwate, Japan, 5 St. Luke's International University, Tokyo, Japan, 6 Morioka Tsunagi Onsen Hospital, Morioka, Iwate, Japan, 7 Faculty of Humanities and Social Sciences, Iwate University, Morioka, Japan, 8 Department of Neurology and Gerontology, Iwate Medical University, Shiwa-gun, Iwate, Japan, 9 Department of Neurosurgery, Iwate Medical University, Shiwa-gun, Iwate, Japan, 10 Division of Nephrology and Hypertension, School of Medicine, Iwate Medical University, Shiwa-gun, Iwate, Japan, 11 Health Service Center, Iwate University, Morioka, Japan, 12 Iwate Health Service Association, Morioka, Iwate, Japan, 13 Department of Nutritional Sciences, Morioka University, Takizawa, Japan, 14 Research Institute of Strategy for Prevention, Tokyo, Japan

* shutakahashi-iwt@umin.ac.jp

**Data Availability Statement:** All relevant data are within the paper and S1–S3 Tables.

## Abstract

### Objectives

The risk factors that contribute to future functional disability after heart failure (HF) are poorly understood. The aim of this study was to determine potential risk factors to future functional disability after HF in the general older adult population in Japan.

### Methods

The subjects who were community-dwelling older adults aged 65 or older without a history of cardiovascular diseases and functional disability were followed in this prospective study for 11 years. Two case groups were determined from the 4,644 subjects: no long-term care insurance (LTCI) after HF (n = 52) and LTCI after HF (n = 44). We selected the controls by randomly matching each case of HF with three of the remaining 4,548 subjects who were event-free during the period: those with no LTCI and no HF with age +/-1 years and of the same sex, control for the no LTCI after HF group (n = 156), and control for the LTCI after HF group (n = 132). HF was diagnosed according to the Framingham diagnostic criteria. Individuals with a functional disability were those who had been newly certified by the LTCI during the observation period. Objective data including blood samples and several socioeconomic items in the baseline survey were assessed using a self-reported questionnaire.

**Funding:** This work was supported by JSPS KAKENHI grant numbers JP17K09126 (KT), JP16KT0009 (KT), and JP20K18858 (ST). This research was also supported by a grant-in-aid from the Ministry of Health, Labour and Welfare, Health and Labour Sciences Research Grants, Japan (Comprehensive Research on Cardiovascular Disease and Life-Related Disease: H23-Junkankitou [Seishuu]-Ippan-005 (SK); H26-Junkankitou [Seisaku]-Ippan-001 (SK), H29-Junkankitou-Ippan-003 (SK), and 20FA1002 (SK)). The funders had no role in study design, data collection and analysis, decision to publish, or preparation of the manuscript.

**Competing interests:** The authors have declared that no competing interests exist.

## Results

Significantly associated risk factors were lower educational levels (odds ratio (OR) [95% confidence interval (CI)]: 3.72 [1.63–8.48]) in the LTCI after HF group and hypertension (2.20 [1.10–4.43]) in no LTCI after HF group. Regular alcohol consumption and unmarried status were marginally significantly associated with LTCI after HF (OR [95% CI]; drinker = 2.69 [0.95–7.66]; P = 0.063; unmarried status = 2.54 [0.91–7.15]; P = 0.076).

## Conclusion

Preventive measures must be taken to protect older adults with unfavorable social factors from disability after HF via a multidisciplinary approach.

## Introduction

A phenomenon that has the potential to become a major health-related issue in developed countries is rapid population aging combined with low birth rates. With the number of older adults aged 65 years or older on the rise [1], most countries will soon have to face several social and economic challenges specific to the health and welfare of this population. Thus, it is imperative for both the government and private sectors to develop new approaches to social security.

In the context of Japan, which has achieved the highest longevity in the world (81 years in men and 87 years in women) [2], the number of older adults aged 65 or older is estimated at 35.6 million, equivalent to 28.1% of the population [3]. Further, the older adult population is expected to reach a peak of 38.9 million in 2042 [4]. In this scenario, the ongoing issue of sustaining older adults' health through social security, complemented by a balanced national budget for pension and healthcare expenditure, is particularly relevant [5].

Heart failure (HF), one of the leading causes of death worldwide, is a rising global health problem. In Japan, where the number of patients with HF is rapidly increasing in tandem with population aging, the number of cases of HF is estimated at one million [6]. Patients with HF typically develop functional disability [7], because of which they have high support needs. Therefore, better management following HF is a pressing matter.

Functional disability can be identified by a decline in activities of daily living (ADL) or the need for admission to a nursing home [8]. In 2000, the Japanese government started a long-term care insurance (LTCI) system for older adults aged 65 and above, the goal of which is to prevent functional disability and aid in daily life management [9]. Under the LTCI system, every Japanese older adult is entitled to certain services according to their functional levels. The LTCI levels were objectively determined by the Nursing Care Needs Certification Board using the data of patients' physical and mental disability with the primary care doctors' opinions. As LTCI levels are an accurate reflection of functional abilities [10], they are used as an outcome of functional disability [11–14], that is, dozens of studies have investigated the association between LTCI as an indicator of functional disability and physical disability/cognitive decline, including frailty and dementia [11, 15–17]. LTCI has also been related to mortality and morbidity in previous studies [18].

The risk factors for HF have been well established in a previous cohort study [19]. Some studies have assessed functional disability using frailty [20, 21], ADL [7], and muscle strength [22] after HF. Studies have identified risk factors for the development of disability in patients with HF [7, 21, 23]. One Japanese study showed that functional disability assessed by LTCI is

related to a high readmission rate after HF [24]. However, these studies evaluated individual characteristics, such as ejection fraction, as prognosis factors. In addition, the participants in these studies were limited to hospitalized patients with HF. Therefore, the premorbid risk factors that contribute to future disability after HF in community-dwelling older adults are poorly understood. In order to identify the impact of HF on functional disability, it is important to determine the potential risk factors for incident functional disability in community-dwelling older adults with and without HF. Recent studies have shown the difference of risk factors between the outcomes for incidence of HF and for prognosis after HF, e.g., "obesity paradox", where obesity is a risk factor for incident HF, but not for prognosis after HF [25]. The identification of risk factors between no functional disability and functional disability after HF would contribute to earlier detection of high-risk patients and timely interventions. Therefore, the aim of this study was to determine the preclinical risk factors for functional disability subsequent to HF in the general older adult population in Japan.

## Materials and methods

The research plan was deliberated and approved by the Ethics Committee of Iwate Medical University Institute Review Board #1 (approval no. H13-33). The rights and welfare of the participants in this study were protected by the ethical guidelines outlined in the Declaration of Helsinki.

### Study population

In the original cohort of the Iwate-KENCO study, participants were recruited from a community-based population living in the Ninohe, Kuji, and Miyako districts of Iwate Prefecture, Japan. The total number of participants who agreed to join the Iwate-KENCO study in the three districts above was 26,469. After excluding participants in Miyako (n = 10,542) and those aged 64 years or younger (n = 8,189), the participants were left 7,738 community-dwelling older adults aged 65 years or older in the Kuji and Ninohe areas (Fig 1). We recruited people who participated in the annual health check-ups of self-employed citizens in the National Healthcare Insurance in Japan from 2002 to 2004. Individuals who agreed to participate in our survey took the baseline survey immediately during the terms. These health check-ups are usually conducted in community centers. The details of the methodology of the Iwate-KENCO

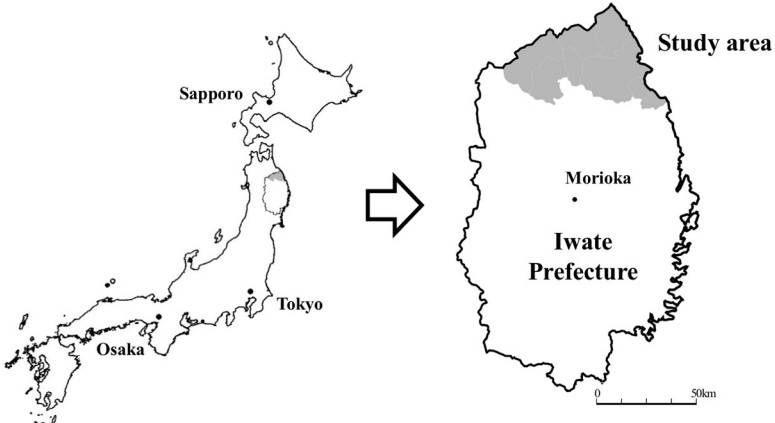

**Fig 1. Map of the survey area.** The figure shows a map of Japan and of Iwate Prefecture. The gray areas describe the study area in northern Iwate. The municipalities included in our study were Ninohe and Kuji.

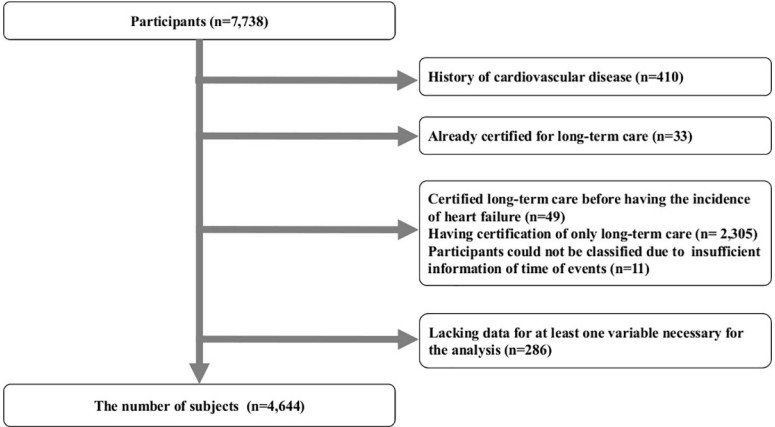

**Fig 2. Flowchart of subject selection.** The number of participants in the original cohort consisted of 7,738 people in the baseline survey. After all the necessary exclusions, 4,644 people remained in the present study.

have been described in a previous paper [26]. We excluded subjects who had a history of stroke, myocardial infarction, or HF (n = 410), those who had already received LTCI (n = 33), those who received LTCI but did not have HF (n = 2,305), those who had already received LTCI prior to HF onset (n = 49), those who could not be classified owing to insufficient information of time of events (n = 11), and those who lacked data for at least one variable that was necessary for analysis (n = 286) (Fig 2). Two case groups were identified from the remaining 4,644 subjects: no LTCI after HF (n = 52) and LTCI after HF (n = 44). Controls were selected by randomly matching each case of HF with three of the remaining 4,548 subjects who were event-free during the follow-up period, such as those with no LTCI and no HF and who were alive on the date of diagnosis of HF for the case with age +/-1 years and same sex; control for no LTCI after HF group (n = 156); and control for LTCI after HF group (n = 132).

## Outcome definition

Individuals with functional disability were defined as those who had been newly certified by the LTCI system during the observation period. We followed the reports about LTCI certification which was assessed and registered by each municipality. Certification of LTCI is determined based on assessment results by the Certification Committee for Long-term Care Need in municipalities based on nationally uniform criteria implemented by the Government of Japan. LTCI levels were determined by qualified personnel using on-site assessment through structured questionnaires and medical interviews to evaluate patients' physical and cognitive abilities [9]. Trained local government officials conducted a home visit to evaluate the patient's nursing care needs using a questionnaire, containing questions regarding the patient's current physical status, mental status (73 items), and medical procedures (12 items) [9]. For example, in the dimension of paralysis and limitation of joint movement, the officials assessed the presence of paralysis or limitations of joint movement in various parts of the body. In terms of functional capacity in LTCI certification, candidates were also assessed for activities of daily living (ADLs) and instrumental ADLs (IADLs), such as dressing and personal hygiene. The results of assessments by government officials were typed into a computer to calculate the applicant's standardized scores for the seven dimensions of physical and mental status, estimate the time taken for the nine categories of care (grooming/bathing, eating, toileting, transferring, eating, assistance with IADL, behavioral problems, rehabilitation, and medical services), and the elderly were assigned a care-needs level based on the total estimated care

minutes. Once approved for LTCI, older adults are eligible to take a monetary amount of services according to their level of disability. The present LTCI certification system comprises seven levels: support levels 1 and 2 (persons who require daily assistance) and care need levels 1 (minimal disability) to 5 (bedridden status and dementia and/or physical impairment). Falling into any of these categories was defined as functional disability at the endpoint.

We also registered patients with HF for hospital inpatients by checking the medical records of the referral hospitals. The definition of HF was based on the Framingham criteria, and patients were identified through regional registration survey data. The registration details have been reported in a previous article [27].

This information was used for both, LTCI certification and the incidence of HF. The monitoring started from 2002 to 2004 and ended on December 31, 2014.

## Measurements

Subjects underwent anthropometric examinations (body weight [kg] and height [cm]) with light clothing and no shoes. Systemic blood pressure was determined in the sitting position using an automatic digital device. Blood pressure measurements were performed twice in the baseline survey, and the mean value was used for statistical analyses.

Blood samples were drawn from a peripheral vein while the subjects were seated. We measured the serum levels of total cholesterol (TC; mg/dL), high-density lipoprotein cholesterol (HDLC; mg/dL), glycosylated hemoglobin (Hb; %), and hemoglobin (hemoglobin A1c (HbA1c); %). Non-high-density lipoprotein cholesterol (non-HDLC; mg/dl) was calculated by subtracting HDLC from TC. The estimated glomerular filtration rate (eGFR; mL/min/1.73 $m^2$) was calculated using the formula derived by the Chronic Kidney Disease Epidemiology Collaboration.

Subjects were administered a self-reported questionnaire covering medical history, including the status of prescribed drugs for hypertension, diabetes mellitus, dyslipidemia, stroke, myocardial infarction, and HF. The questionnaire also covered lifestyle factors such as smoking status (current smoker or nonsmoker) and alcohol consumption (nondrinker or regular drinker; drinking alcohol more than five times per day). With regard to cardiovascular risk factors, hypertension was ascertained by a systolic blood pressure $\geq$ 140 mmHg and/or diastolic blood pressure $\geq$ 90 mmHg and/or the use of antihypertensive agents. Diabetes mellitus was defined as a non-fasting glucose concentration $\geq$ 200 mg/dL, and/or fasting blood glucose level $\geq$ 126 mg/dL, and/or hemoglobin A1$_c$ value $\geq$ 6.5%, and/or the use of antidiabetic medicines, including insulin. Dyslipidemia was ascertained by a serum TC $\geq$ 220 mg/dL, serum HDLC < 40 mg/dL, and/or the use of antilipidemic agents. Regarding social factors, marital status was classified into two groups: married or unmarried. Educational attainment was classified into two categories according to duration: low ($\leq$ 6 years) and high ($\geq$ 7 years). Job status was categorized into two groups (unemployed or employed). The details of this process have been described in a previous article [28].

## Statistical analyses

All statistical analyses were stratified on the basis of whether or not subjects received LTCI after HF. The characteristics of cases and controls were compared using the Student's t- test (continuous variables) and the chi-squared test (categorical variables). A conditional logistic regression analysis was used to calculate the odds ratios (ORs) and 95% confidence intervals (CIs) of the risks of no LTCI after HF and LTCI after HF. We built multivariable models that sequentially introduced a range of confounding variables: Model 1, adjustment for lifestyle; Model 2, addition of cardiovascular risk factors; and Model 3, addition of social factors. To

avoid the influence of potential pre-existing HF at baseline, we performed a similar analysis after excluding HF cases within two years from the baseline survey.

All-cause deaths and migration were confirmed by the official resident registration data issued by the local government offices (December 2009). All P-values were based on two-sided tests, and P-values < 0.05 were considered statistically significant. SPSS version 24.0 (IBM Corp., Armonk, NY, USA) was used for all statistical analyses.

## Results

The median follow-up period was 10.9 years for a total of 4,193 person-years. The median period from the baseline to the incidence of HF was 5.5 years (interquartile range, 2.4–8.6 years) in the group with no LTCI after HF, and LTCI after HF. The median period from the incidence of HF to LTCI certification was 1.3 years (interquartile range, 0.3–4.7. years) in the LTCI after HF. Table 1 presents the comparison of baseline characteristics between the case and control groups. The mean age and sex distribution were almost the same between cases and controls in the no LTCI after HF and LTCI after HF groups.

**Table 1. Baseline characteristics.**

|  | No LTCI after HF | | | LTCI after HF | | |
|---|---|---|---|---|---|---|
|  | Control (n = 156) | Case (n = 52) | p-values | Control (n = 132) | Case (n = 44) | p-values |
|  | Mean (SD)/ n (%) | Mean (SD)/ n (%) |  | Mean (SD)/ n (%) | Mean (SD)/ n (%) |  |
| Age (yrs) | 72.1 (4.2) | 72.3 (4.3) | 0.799 | 74.5 (4.5) | 74.6 (4.7) | 0.939 |
| BMI (kg/m$^2$) | 23.6 (3.8) | 23.7 (2.7) | 0.822 | 23.5 (3.3) | 24.0 (3.8) | 0.408 |
| SBP (mmHg) | 130.5 (19.7) | 137.2 (20.3) | 0.035 | 132.2 (18.4) | 137.2 (21.3) | 0.136 |
| DBP (mmHg) | 76.9 (11.4) | 74.9 (10.7) | 0.255 | 75.1 (9.8) | 75.0 (10.8) | 0.940 |
| TC (mg/dl) | 198.5 (29.0) | 192.1 (34.7) | 0.186 | 195.5 (33.0) | 188.3 (32.2) | 0.210 |
| HDLC (mg/dl) | 57.0 (14.2) | 60.5 (17.8) | 0.198 | 56.6 (15.2) | 55.7 (14.3) | 0.739 |
| Non-HDLC (mg/dl) | 141.6 (28.7) | 131.6 (34.1) | 0.039 | 138.9 (30.9) | 132.6 (32.9) | 0.250 |
| Hb (g/dl) | 13.7 (1.4) | 13.8 (1.3) | 0.870 | 13.6 (1.4) | 13.4 (1.4) | 0.605 |
| HbA1c (%) | 5.5 (0.6) | 5.5 (0.8) | 0.767 | 5.6 (1.0) | 5.5 (0.5) | 0.469 |
| eGFR (mL/min/1.73 m$^2$) | 71.5 (7.0) | 69.2 (9.1) | 0.112 | 70.4 (7.2) | 69.9 (9.7) | 0.719 |
| Sex (men) | 87 (55.8) | 29 (55.8) | 1.000 | 66 (50.0) | 22 (50.0) | 1.000 |
| Age group (65–69) | 39 (25.0) | 12 (23.1) | 0.994 | 18 (13.6) | 6 (13.6) | 1.000 |
| (70–74) | 68 (43.6) | 23 (44.2) |  | 48 (36.4) | 16 (36.4) |  |
| (75–79) | 43 (27.6) | 15 (28.8) |  | 45 (34.1) | 15 (34.1) |  |
| (≥ 80) | 6 (3.8) | 2 (200.0) |  | 21 (15.9) | 7 (15.9) |  |
| Current smoker | 28 (17.9) | 8 (15.4) | 0.672 | 14 (10.6) | 4 (9.1) | 1.000 |
| Regular drinker | 37 (23.7) | 10 (19.2) | 0.503 | 23 (17.4) | 14 (31.8) | 0.042 |
| Hypertension | 75 (48.1) | 34 (65.4) | 0.030 | 71 (53.8) | 26 (59.1) | 0.540 |
| Diabetes mellitus | 9 (5.8) | 6 (11.5) | 0.164 | 10 (7.6) | 2 (4.5) | 0.733 |
| Dyslipidemia | 54 (34.6) | 14 (26.9) | 0.306 | 43 (32.6) | 11 (25.0) | 0.345 |
| Unmarried status | 35 (22.4) | 13 (25.0) | 0.704 | 35 (26.5) | 18 (40.9) | 0.071 |
| Lower educational level | 44 (28.2) | 14 (26.9) | 0.858 | 35 (26.5) | 23 (52.3) | 0.002 |
| Unemployed status | 67 (42.9) | 16 (30.8) | 0.120 | 38 (28.8) | 16 (36.4) | 0.345 |

Continuous variables are expressed as mean (standard deviation) using the Student's t test, and categorical variables were calculated as the number of cases (proportion, %) using the chi-squared test.

LTCI, long-term care insurance; HF, heart failure; BMI, body mass index; SBP, systolic blood pressure; DBP, diastolic blood pressure; TC, total cholesterol; HDLC, high-density lipoprotein cholesterol; non-HDLC, non-high-density lipoprotein cholesterol; Hb, blood hemoglobin; HbA1c, glycosylated hemoglobin; eGFR, estimated glomerular filtration rate; SD, standard deviation.

In the no LTCI after HF group, while the percentage of hypertension and systolic blood pressure levels were significantly higher in cases than controls, the value of non-HDLC was significantly lower in cases than controls. In the LTCI after HF group, although the proportion of regular drinkers was significantly higher in cases than controls, there were no significant differences in the proportions of cardiovascular risk factors and social factors between the two groups.

Table 2 shows the ORs for the no LTCI after HF and LTCI after HF groups using conditional logistic regression analyses. In Model 2 for no LTCI after HF (adjustment for lifestyle and cardiovascular risk factors), ORs of cases were significantly higher in subjects with hypertension. This association remained significant after adjustment for social factors (OR [95% CI]; hypertension = 2.20 [1.10–4.43]). In Model 1 for LTCI after HF (adjustment for smoking and drinking status), ORs of cases were significantly higher in regular drinker. This significant association between regular drinking and LTCI after HF disappeared after adjustment for cardiovascular risk factors (Model 2). In Model 3 for LTCI after HF (adjustment for social factors), ORs were significantly higher for cases with lower educational levels (OR [95% CI]; low educational levels = 3.72 [1.63–8.48]). Fig 3 shows the ORs for each risk factor in the no LTCI after HF and LTCI after HF groups. Regular drinking and unmarried status were marginally significantly associated with LTCI after HF (OR [95% CI]; regular drinking = 2.69 [0.95–7.66]; P = 0.063; unmarried status = 2.54 [0.91–7.15]; P = 0.076).

In the sensitivity analysis, while hypertension's significant association with no LTCI after HF disappeared, the association with LTCI after HF remained significant for those with lower educational levels after excluding sub-clinical cases of HF (S1 Table).

## Discussion

The present study demonstrated that LTCI after HF was associated with lower educational levels at baseline in the general population, even after adjusting for lifestyle and established cardiovascular risk factors. No LTCI after HF was associated with hypertension. We found that the premorbid risk factors differed between in the no LTCI after HF and LTCI after HF groups.

Previous studies have shown some predictors of functional decline after HF [7, 20–22, 24]. While those studies examined risk factors for functional disability, their timing of assessment differed from our study, that is, while previous studies were conducted on hospital admission

**Table 2. Odds ratios for the categories of no long-term care insurance after heart failure and long-term care insurance after heart failure using conditional logistic regression analyses.**

| | No LTCI after HF | | | LTCI after HF | | |
|---|---|---|---|---|---|---|
| | Model 1 | Model 2 | Model 3 | Model 1 | Model 2 | Model 3 |
| | OR (95% CI) | OR (95% CI) | OR (95% CI) | OR (95% CI) | OR (95% CI) | OR (95% CI) |
| **Current smoker** | 0.83 (0.33–2.04) | 0.71 (0.27–1.86) | 0.75 (0.29–1.96) | 0.65 (0.19–2.17) | 0.73 (0.21–2.47) | 0.52 (0.15–1.85) |
| **Regular drinker** | 0.74 (0.32–1.71) | 0.78 (0.32–1.91) | 0.89 (0.35–2.23) | 2.80 (1.14–6.89) | 2.51 (0.99–6.41) | 2.69 (0.95–7.66) |
| **Hypertension** | | 2.15 (1.07–4.31) | 2.20 (1.10–4.43) | | 1.15 (0.55–2.38) | 1.13 (0.49–2.57) |
| **Diabetes mellitus** | | 2.04 (0.69–6.01) | 1.95 (0.65–5.86) | | 0.57 (0.12–2.84) | 0.52 (0.10–2.78) |
| **Dyslipidemia** | | 0.71 (0.35–1.43) | 0.69 (0.34–1.40) | | 0.72 (0.31–1.66) | 0.59 (0.23–1.53) |
| **Unmarried status** | | | 1.05 (0.46–2.42) | | | 2.54 (0.91–7.15) |
| **Lower educational level** | | | 1.00 (0.45–2.21) | | | 3.72 (1.63–8.48) |
| **Unemployed status** | | | 1.63 (0.83–3.23) | | | 0.67 (0.25–1.82) |

LTCI, long-term care insurance; HF, heart failure; OR, odds ratio; CI, confidence interval.

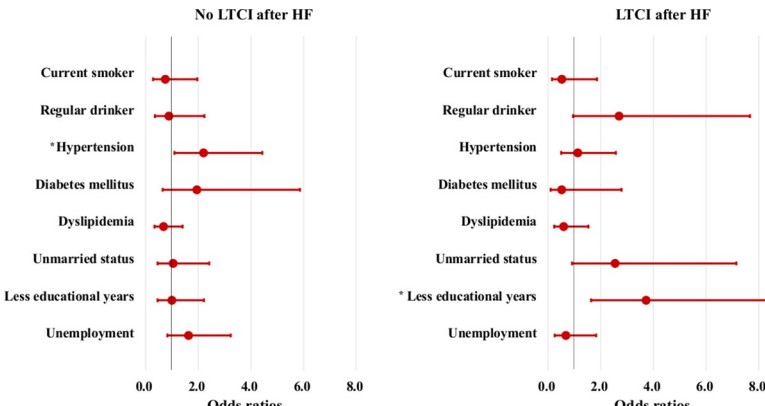

**Fig 3. Odds ratios (95% confidence interval) for having each risk factor in the no long-term care insurance after heart failure and long-term care insurance after heart failure groups.** Error bars represent the 95% confidence intervals. Adjustment for smoking, drinking, hypertension, diabetes mellitus, dyslipidemia, unmarried status, lower educational level (< 7 years), and unemployed status using conditional logistic regression analysis. LTCI = long-term care insurance, HF = heart failure. *Statistically significant (P < 0.05).

and/or at discharge, we gathered information on risk factors in subjects initially free of diseases and identified etiologic precursors by following up for 11 years. Therefore, the items evaluated in previous studies are different from those in our study. Existing research has assessed socio-economic status and cardiovascular risk factors at the time of hospital admission for HF. In particular, studies have focused on cardiac function after HF, for example, ejection fraction and valvular diseases [24, 29]. In the present study, we assessed baseline risk factors obtained several years before HF onset.

In comparison to our results, previous studies have demonstrated a significantly higher incidence of functional disability after HF [7, 20–22]. Although the areas of focus in these studies were similar to the current study in terms of examining the association of particular variables with functional disability after HF, their endpoints—frailty, ADL, or muscle weakness—differed. In our study, functional disability as the endpoint was defined as any levels of LTCI (support level 1 or more), which has been validated in a previous study [30]. Support level 1 indicates persons who have limitations in instrumental ADL but are independent in basic ADL. To the best of our knowledge, no studies have demonstrated the risk factors for functional disability, including instrumental limitations, following HF. Those with lower educational levels were found to be at risk even in the case of a trivial functional disability resulting from HF.

Low educational level was the main predictor of functional disability among patients with LTCI after HF. Although we could not elucidate the precise mechanism of the association between educational level and LTCI after HF, there are a few possible explanations. First, there might be a difference in physical function in people in the LTCI after HF. A previous systemic review has indicated an association between lower socioeconomic status, including low educational level, and a higher incidence of HF [31]. A lower educational level in people with HF is related to a lower level of functional disability assessed ADL with emotional distress, poorer general health, and more anxiety [32]. In contrast, patients with higher educational levels show long-term improvements in functional limitations related to emotional problems [33]. As LTCI has been linked to both physical dysfunction and cognitive decline [12, 14, 16, 34], in patients with HF, the combination of low educational attainment and poor emotional status might lead to the development of functional disability. Second, access to medical doctors might differ in patients with lower educational levels. A study has shown differences in

physicians' care by educational levels among patients with congestive HF [35]. Barriers to appropriate care might impact patient outcomes. Third, it is assumed that compliance with appropriate therapy and rehabilitation after HF have an influence on future LTCI requirement. The major cause of readmission in older adults with HF is inappropriate self-care in daily life [36]. To delay the progression of HF, better compliance with optimal treatment is required [37, 38]. Patients need to understand and adhere to appropriate treatments and rehabilitation to prevent HF progression. However, older adults with poor educational attainment might not display good compliance with chronic disease control programs [39]. For example, older adults with HF might not take medications or dietary guidance appropriately. Further, even though older adults tend to receive drugs for other comorbidities, they might suspend effective cardiac rehabilitation. Difficulties in self-care lead to loss of independence and consequently, a lower quality of life [29]. In addition, reverse causation might occur; decreased cognitive functioning, which is also one of the major reasons for receiving LTCI, contributes to poor adherence to therapeutic regimens [40]. As patients with HF with poor social factors continue to display poor compliance over the long term, they may experience progression of HF, which further deteriorates the prognosis of functional disability.

Recent studies have shown the difference of risk factors between outcomes for incidence of HF and for prognosis after HF [25]. Accumulating evidence accounts for the difference between the two. Some studies have shown that hypertension increases the risk of HF [19, 41]. In contrast, one study that examined long-term outcomes after HF showed that hypertension decreased mortality, although all-cause readmission increased after HF [42]. In the present study, hypertension was related only to no LTCI after HF. We could interpret our results that hyperextension had a high predictive value for the incidence of HF, which has already been described as an established risk factor for HF. In contrast, although we could not determine hypertension as a protective effect against functional disability among patients with HF due to no significant impact on LTCI after HF, the finding that hypertension did not have a significant association with LTCI after HF suggests that social factors, including educational level, have a substantial impact on future functional disability via HF compared with established cardiovascular risk factors. Higher ORs were found in unmarried people with LTCI after HF, but the difference was not statistically significant. Previous studies have shown that people who live alone and are unmarried have a significantly higher risk of readmission after HF [7, 43]. Essentially, the absence of family support is related to functional decline [13]. We expected that loss of independence with no support from close relatives would have a strong influence on functional decline. We found that classic HF prognostic factors, including hypertension, were a substantial indicator of patients with no LTCI after HF, but they did not have a significant association with LTCI after HF. Although classic HF factors are treatable, educational level and marital status cannot be modified through interventions by healthcare workers. Our findings provide new insight into strategies for the prevention of functional disability after HF.

In addition, we observed a marginally significant association between regular drinking and LTCI after HF. While heavy alcohol consumption is recognized as a heightened risk factor for HF because of physiological damage [38], prospective studies have failed to reach a consensus about the association between excessive alcohol consumption and LTCI [11]. Nevertheless, unmarried status, lower educational attainment, and excessive drinking might be regarded as proxies for unfavorable social risk factors that strongly contribute to the future risk of LTCI subsequent to HF; however, they did not contribute to the risk of incident HF without LTCI. Therefore, to prevent older adults from experiencing future functional disability, there is a need for measures that take these proxies into consideration.

The present study has several limitations. First, we did not fully investigate the association between functional disability and HF adjusting for relevant risk factors related to either HF or

LTCI, for example, low body mass index or poor nutrition. Owing to the low number of cases of HF (n = 96), we had to limit the covariates when considering appropriate statistical analysis. Second, some patients with HF who were treated in outpatient departments were not registered as cases of HF because they did not adequately fulfill the Framingham diagnostic criteria, which might have led to the underestimation of our results. Third, our study did not include some important confounding factors, such as household equivalent income, because they were not featured in the cohort data. Detailed information on income could have been helpful in the elucidation of a more precise association between educational level and functional disability after HF. Fourth, details of functional disability and HF could not be determined, including physical disability or cognitive dysfunction in LTCI, and HF severity, such as ejection fraction in the left ventricle or NYHA in HF. This information would have enabled us to determine the details of the mechanisms. Fifth, although LTCI is neither a HF specific assessment tool for functional disability, nor validated by some physiological index such as maximum oxygen uptake in exercise, previous studies have examined the external validity of LTCI in physical and cognitive disabilities in elderly individual; a study determined that the levels of LTCI certification are well associated with the ability to perform activities of daily living, assessed by the Barthel Index (Spearman's coefficient = −0.86) and the Mini-Mental State Examination scores (Spearman's coefficient = −0.42) [30]. Nevertheless, a comparison between LTCI and other scales for patients after HF might provide precise information on post HF-specific functional disability, for example, physical disability caused by dyspnea on exertion. Further studies are therefore needed. Sixth, the results have limited generalizability because the subjects were younger than those living in the same areas in the 2000 census (average age of citizens in 2005 = 75.0 years). It is assumed that individuals in the present study have better function and a lower incidence of HF. This should be taken into consideration when we generalize our results. Finally, participants in the present study were those who could access the places where our survey was conducted by themselves. These centers were located in each elementary school district; they were close to houses because these community centers were built for citizens who can access them easily by walking. Subjects had relatively good physical function because they were able to travel to the survey venues. Owing to this selection bias, the possibility of underestimation of the true incidence of disability in the general population cannot be eliminated.

## Conclusions

In this study, we used the premorbid data for community-dwelling older adults and determined the existing risk factors for future functional disability after HF through an 11-year follow-up. Lower educational levels were seen to have an impact on the development of functional disability after HF. Patients with HF are required to manage follow-up treatments to prevent further functional disability. Our findings suggest the need for a multidisciplinary approach to long-term monitoring of treatment adherence and lifestyle management in these individuals.

## Supporting information

**S1 Table. Odds ratios (95% confidence interval) for the categories of no long-term care insurance after heart failure and long-term care insurance after heart failure excluding subjects with sub-clinical.**
(DOCX)

**S2 Table. Data in the present study.**
(PDF)

**S3 Table. Explanation of variables.**
(PDF)

## Acknowledgments

The authors would like to thank the participants of this study, the staff of the Iwate Health Service Association, and the staff in all municipalities (Iwate Prefecture, Ninohe City, Ichinohe Town, Karumai Town, Kunohe Village, Yamada Town, Miyako City, Iwaizumi Town, Tanohata Village, Kuji City, Fudai Village, Noda Village, and Hirono Town). This project was conducted with the support of the Takemi Program in International Health at the Harvard T.H. Chan School of Public Health. We thank Prof. Ichiro Kawachi for supervising the lead author during his fellowship.

## Author Contributions

**Conceptualization:** Shuko Takahashi, Kozo Tanno.

**Data curation:** Yuki Yonekura.

**Formal analysis:** Shuko Takahashi.

**Funding acquisition:** Kiyomi Sakata.

**Methodology:** Kozo Tanno.

**Resources:** Yasuhiro Ishibashi, Shinichi Omama, Fumitaka Tanaka.

**Supervision:** Yuki Yonekura, Masaki Ohsawa, Toru Kuribayashi, Toshiyuki Onoda, Kiyomi Sakata.

**Writing – original draft:** Shuko Takahashi.

**Writing – review & editing:** Kozo Tanno, Yuki Yonekura, Masaki Ohsawa, Toru Kuribayashi, Yasuhiro Ishibashi, Shinichi Omama, Fumitaka Tanaka, Toshiyuki Onoda, Kiyomi Sakata, Makoto Koshiyama, Kazuyoshi Itai, Akira Okayama.

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
