## [Decision Letter · Decision Letter 0]

23 Mar 2021

PONE-D-21-00523

Low educational level increases functional disability risk subsequent to heart failure in Japan: on behalf of the Iwate KENCO study group

PLOS ONE

Dear Dr. Takahashi,

Thank you for submitting your manuscript to PLOS ONE. After careful consideration, we feel that it has merit but does not fully meet PLOS ONE’s publication criteria as it currently stands. Therefore, we invite you to submit a revised version of the manuscript that addresses the points raised during the review process.

We look forward to receiving your revised manuscript.

Kind regards,

Antonio Palazón-Bru, PhD

Academic Editor

PLOS ONE

Journal Requirements:

2. In the Methods please provide a justification for the stratification of educational level according to the years of education.

3. We note that Figure 1 in your submission contains map images which may be copyrighted.

We require you to either (a) present written permission from the copyright holder to publish this figure specifically under the CC BY 4.0 license, or (b) remove the figure from your submission:

b. If you are unable to obtain permission from the original copyright holder to publish this figure under the CC BY 4.0 license or if the copyright holder’s requirements are incompatible with the CC BY 4.0 license, please either i) remove the figure or ii) supply a replacement figure that complies with the CC BY 4.0 license. Please check copyright information on all replacement figures and update the figure caption with source information. If applicable, please specify in the figure caption text when a figure is similar but not identical to the original image and is therefore for illustrative purposes only.

Reviewers' comments:

Reviewer's Responses to Questions

**Comments to the Author**

1. Is the manuscript technically sound, and do the data support the conclusions?

Reviewer #1: Yes

Reviewer #2: Partly

Reviewer #3: Partly

2. Has the statistical analysis been performed appropriately and rigorously? 

Reviewer #1: Yes

Reviewer #2: Yes

Reviewer #3: Yes

3. Have the authors made all data underlying the findings in their manuscript fully available?

Reviewer #1: Yes

Reviewer #2: Yes

Reviewer #3: Yes

4. Is the manuscript presented in an intelligible fashion and written in standard English?

Reviewer #1: Yes

Reviewer #2: Yes

Reviewer #3: Yes

5. Review Comments to the Author

Reviewer #1: Please summarize concisely the principal implications of the finding. Explain how the findings may be important for policy, practice, or research….

Please provide recommendations for further research.

Please explain how the results and conclusions of this study are important.

Reviewer #2: Cohort study in which researchers identify patients who start out with heart failure and follow them for 11 years with the aim of identifying risk factors for the development of disability.

Disability in patients with HF is common. The study is interesting because it is a prospective study in community-dwelling older adults aged 65 or older.

The study analyzed two groups of patients with HF, one of them without disability (no LTCI) and the other one with disability (LTCI). Taking into account the objective of the study, it is not understood why the group of non-LTCI patients is analyzed. Comparison of the non-LTCI after HF group with the non-LTCI non-HF group identifies hypertension as a risk factor. In this case, it would be a risk factor for developing HF, fact already known.

In the study, the functional disability is determined by the LTCI certification. This certification is done considering two dimensions: A) physical and psychological status and B) use of medical procedures. Its objective is to recognize the right to social and health benefits. Considering that, from the clinical point of view, it is not the usual way of assessing functional capacity, in the methods section, to facilitate the reader's understanding, it should be explained in a summarized way how the functional capacity of these patients is assessed.

In the LTCI group, the proportion of patients with a lower educational level in the cases is double that in the controls. In these patients, the severity of heart failure is also probably higher than in highly educated patients. To control this possible bias, the model should have been adjusted with some variable indicative of heart failure severity (LVEF or NYHA).

The LTCI system has been developed in Japan in order to assess care-needs in persons aged 65 and older. In order to understand the external validity of the study results, it is necessary that in the discussion section the results should be assessed in comparison with other scales to measure disability, generic or specific for patients with HF, such as the MLHFQ.

The paragraph on page 5, lines 100-103, is repeated on page 7 lines 128-130.

References must include studies that have identified risk factors for the development of disability in patients with heart failure.

Reviewer #3: Summary of the Research and Overall Impression

This paper uses a novel dataset to examine the pre-morbid risk factors that contribute to future disability (defined by enrollment in long-term care insurance, or LTCI) after heart failure (HF) in community-dwelling older adults in Japan. The goal is to contribute to earlier detection of high-risk patients and thereby facilitate more timely interventions. From a group of nearly 8,000 patients, the authors identified 96 who developed heart failure over the study period, and split these into two case groups, one of which required LTCI after HF, and the other which did not. Both groups were matched 3-to-one to same age/sex controls who had neither HF nor LTCI during the study period, and cases were compared to controls in both groups. The main results are that hypertension is associated with HF without LTCI, and lower education levels are associated with HF with LTCI.

The topic is an important one given the aging population in Japan, the increasing prevalence of heart failure and the caregiving and quality-of-life costs associated with functional disability. However, the paper does not sufficiently explain the study sample or provide enough information about how to interpret these results with confidence.

Specific Areas for Improvement

Major Issues

• More discussion of the case-control study design and how this affects how we interpret the results is needed to understand the study’s implications. For example, the case-control study design for the HF-Non LTCI subgroup needs more explanation. The control group has neither HF nor LTCI, while the case group has both; the comparison between them thus speaks to which pre-morbid conditions contribute to the compound outcome of a diagnosis of HF followed by no functional disability. One might suppose that the Table 2 result that hypertension is positively associated with the compound outcome (HF without LTCI) is picking up the association between hypertension and HF. But if this is so, then the same association should be seen in the (HF *with* LTCI) analysis. What does it mean that no such association appears in this second group? How should one interpret these two results together—that hypertension is protective against functional disability among patients with HF? More discussion about how to interpret this and other results in the paper is needed.

• More detail is needed on the timing of the study and data collection for the reader to follow what has been done. The authors describe the study sample selection process beginning with 7,738 subjects age 65+, and refers to an earlier paper (Ohsawa et al. 2009) for details of the Iwate-KENCO study population. Ohsawa et al. state that the study began in 2002, but it appears that there were many more study participants age 65+ in 2002 (Table 1 in Ohsawa indicates more than 17,000 participants over age 60). To get to the 7,738 starting subjects for the current study, it seems many participants have been excluded from the original study, but there is insufficient detail to understand why that is.

• The prior point is important because, after all sample selection steps, the current study ends up with fewer than 100 subjects (out of 7,738) who develop heart failure over the 11-year study period; this subgroup is further divided into 52 who did not require long-term care insurance and 44 who did. Some discussion of whether these sample sizes are large enough to detect the kinds of relationships we are looking for is needed. Are the authors confident that the relatively few statistically significant results found are not an artifact of too-small sample sizes?

• In the Limitations section, the final limitation mentioned is that patients were required to travel to survey venues to participate in the study. This seems especially troubling as it likely limits the degree of disability among the study population, and makes it even more important to describe the study requirements more fully: what exactly was required of patients to participate in the study (frequency of contact, travel requirements such as distance and mode of transport, whether there were supplementary phone or at-home visits etc.)? Without understanding these items, it is difficult to judge how big an impact this limitation might have had.

Minor Issues

• The current study also says that subjects were followed for 11 years, but does not describe when the monitoring began and ended, nor how frequently patients were surveyed.

• Line 145 suggests that anthropometric measurements were taken repeatedly. If there were multiple regular (say annual) assessments, how were they aggregated over the 11 years of the study?

Miscellaneous Remarks

• Functional disability was coded as a binary variable—either subjects were enrolled in the LTCI program or not. But apparently the LTCI certification system codes seven different levels of disability. Did the authors consider exploring a “dose effect” – that is, do patients with lower educational levels (or who drink more or smoke more) have higher levels of disability? I wonder if this might sharpen the results.

6. PLOS authors have the option to publish the peer review history of their article (what does this mean?). If published, this will include your full peer review and any attached files.

Reviewer #1: No

Reviewer #2: No

Reviewer #3: No

---

## [Author Response · Author response to Decision Letter 0]

30 Apr 2021

Response to reviewers

Title “Low educational level increases functional disability risk subsequent to heart failure in Japan: on behalf of the Iwate KENCO study group” 

PONE-D-21-00523

Dear Editors,

We have revised the manuscript according to the reviewers’ comments. We have added information to the table and text in the revised manuscript according to the reviewers’ comments.

Please review our revised manuscript and our responses to the reviewers’ comments. I hope that the revisions in the manuscript adequately addressed the comments. The responses to the reviewers’ comments have been outlined below.

1) We note that Figure 1 in your submission contains map images which may be copyrighted.

Our response: 

 We created the Map (Figure 1) using the GIS free software named “MANDARA”. That is offered by Kenji Tani in Human Geography, Faculty of Education, Saitama University, Japan (http://ktgis.net/lab/index.php). He described that the map drawn by the MANDARA, it does not matter even if we have to use for any purpose such as commercial use and academic use. The copyright belongs to us. We also confirmed about the copyright to Kenji Tani by e-mail, he responded that the license is not required and we can use freely.Because we created this map (Figure 1) with MANDARA, a copyright holder is me. We did not reprint it from anywhere. Therefore we think that the protection about the copyright does not have any problem.

Reviewer: 1

1. Please summarize concisely the principal implications of the finding.

Our response:

Thank you for reviewing our manuscript. We determined the potential risk factors for future functional disability after heart failure (HF) in the general older adult population in Japan. Functional disability after HF was associated with lower educational levels at baseline, while no functional disability after HF was associated with hypertension through an 11-year follow-up.

2. Explain how the findings may be important for policy, practice, or research….

Our response:

Our findings in the present study were important for policy in the field of cardiovascular prevention. We determined that a lower level of education in the premorbid condition was linked to future functional disability after HF, i.e., we found the target population once people suffered from HF. Health care workers, including officials in the local government, might implement preventive measures for people with a lower level of education. In addition, we should focus on intervening high-risk people in order to maintain their function; for example, people with a lower level of education should receive extensive explanation to adhere to medications appropriately after HF by pharmacists. 

3. Please provide recommendations for further research.

Our response:

We could not combine factors related to functional disability after HF, such as low body mass index or poor nutrition. Moreover, our study did not include some important confounding factors, such as household equivalent income. Future studies are required to adjust for these factors. In addition, because the sample size in the present study was small, a study with a larger sample size may determine a more precise association between functional disability after HF and premorbid risk factors.

4. Please explain how the results and conclusions of this study are important.

Our response:

Individuals with functional disability were defined as those who had been newly certified by the long-term care insurance (LTCI) system. The present study demonstrated that LTCI after HF was associated with lower educational levels at baseline in the general population. However, LTCI after HF was not associated with hypertension. Our findings suggest the need for a multidisciplinary approach to the long-term monitoring of treatment adherence and lifestyle management in these individuals.

 

Reviewer: 2

Cohort study in which researchers identify patients who start out with heart failure and follow them for 11 years with the aim of identifying risk factors for the development of disability.

Disability in patients with HF is common. The study is interesting because it is a prospective study in community-dwelling older adults aged 65 or older.

1. The study analyzed two groups of patients with HF, one of them without disability (no LTCI) and the other one with disability (LTCI). Taking into account the objective of the study, it is not understood why the group of non-LTCI patients is analyzed. Comparison of the non-LTCI after HF group with the non-LTCI non-HF group identifies hypertension as a risk factor. In this case, it would be a risk factor for developing HF, fact already known.

Our response:

Thank you for reviewing our manuscript. As you pointed out, the comparison between the group with no LTCI after HF and the group with LTCI after HF showed the findings that already exist, i.e., odds ratios (ORs) of cases were significantly higher in subjects with hypertension. On the other hand, recent studies have shown the difference in risk factors between outcomes for incident HF and prognosis after HF. For example, obesity is a well-known risk factor for HF. Obesity increases the risk for HF (1) but decreases outcomes after HF such as all-cause death or re-hospitalization (2, 3). Researchers have recognized the “obesity paradox,” in which it is a risk factor for incident HF, but not for prognosis after HF (4). Accumulating evidence accounts for the difference between the two. Based on these studies, we need to examine the difference in these risk factors, i.e., the incidence of HF (the group with no LTCI after HF) and prognosis after HF (the group with LTCI after HF). We found that premorbid risk factors differed between the groups with no LTCI and with LTCI after HF. The important risk factors for LTCI after HF were socioeconomic factors, instead of established cardiovascular risk factors.

 In accordance with the reviewer’s point, we have added text to the Introduction section as follows (page 5, line 95 to line 100):

 “Recent studies have shown the difference of risk factors between the outcomes for incidence of HF and for prognosis after HF, e.g., “obesity paradox”, where obesity is a risk factor for incident HF, but not for prognosis after HF.(4) The identification of risk factors between no functional disability and functional disability after HF would contribute to earlier detection of high-risk patients and timely interventions.”

2. In the study, the functional disability is determined by the LTCI certification. This certification is done considering two dimensions: A) physical and psychological status and B) use of medical procedures. Its objective is to recognize the right to social and health benefits. Considering that, from the clinical point of view, it is not the usual way of assessing functional capacity, in the methods section, to facilitate the reader's understanding, it should be explained in a summarized way how the functional capacity of these patients is assessed.

Our response:

Trained local government officials conducted a home visit to evaluate the patient’s nursing care needs using a questionnaire, containing questions regarding the patient’s current physical status, mental status (73 items), and medical procedures (12 items) (5) (R1 Table 1). For example, in the dimension of paralysis and limitation of joint movement, the officials assessed the presence of paralysis or limitations of joint movement in various parts of the body. In terms of functional capacity in LTCI certification, candidates were also assessed for activities of daily living (ADLs) and instrumental ADLs (IADLs), such as dressing and personal hygiene. The results of assessments by government officials were typed into a computer to calculate the applicant’s standardized scores for the seven dimensions of physical and mental status, estimate the time taken for the nine categories of care (grooming/bathing, eating, toileting, transferring, eating, assistance with IADL, behavioral problems, rehabilitation, and medical services), and the elderly were assigned a care-needs level based on the total estimated care minutes.

In accordance with the reviewer’s suggestion, we have added some sentences to the Materials and Methods section (page 7, line 148 to page 8, line 160). 

“Trained local government officials conducted a home visit to evaluate the patient’s nursing care needs using a questionnaire, containing questions regarding the patient’s current physical status, mental status (73 items), and medical procedures (12 items). (5) For example, in the dimension of paralysis and limitation of joint movement, the officials assessed the presence of paralysis or limitations of joint movement in various parts of the body. In terms of functional capacity in LTCI certification, candidates were also assessed for activities of daily living (ADLs) and instrumental ADLs (IADLs), such as dressing and personal hygiene. The results of assessments by government officials were typed into a computer to calculate the applicant’s standardized scores for the seven dimensions of physical and mental status, estimate the time taken for the nine categories of care (grooming/bathing, eating, toileting, transferring, eating, assistance with IADL, behavioral problems, rehabilitation, and medical services), and the elderly were assigned a care-needs level based on the total estimated care minutes.” 

R1 Table 1. ABOUT HERE

3. In the LTCI group, the proportion of patients with a lower educational level in the cases is double that in the controls. In these patients, the severity of heart failure is also probably higher than in highly educated patients. To control this possible bias, the model should have been adjusted with some variable indicative of heart failure severity (LVEF or NYHA).

Our response:

We agree with your suggestion. The analysis should be adjusted for related variables. However, we did not assess such variables, including LVEF or NYHA, in the present study. We would have determined more precise mechanisms between lower education and functional disability with HF if we had checked HF severity.

Therefore, we have added this information to the Limitation section (page 5, line 231 to page 5, line 233).

 “Fourth, details of functional disability and HF could not be determined, including physical disability or cognitive dysfunction in LTCI, and HF severity, such as ejection fraction in the left ventricle or NYHA in HF.”

4. The LTCI system has been developed in Japan in order to assess care-needs in persons aged 65 and older. In order to understand the external validity of the study results, it is necessary that in the discussion section the results should be assessed in comparison with other scales to measure disability, generic or specific for patients with HF, such as the MLHFQ.

Our response:

There are no validated studies of LTCI compared with other scales of disability in elderly patients with HF. The Minnesota Living with Heart Failure Questionnaire is a health-related quality of life questionnaire for patients post HF, which was validated by assessing cardiopulmonary exercise tolerance test. Neither is LTCI a disease-specific assessment tool for functional disability, nor is it validated by physiological indices such as maximum oxygen uptake during exercise. However, even though there might be no disease-specific assessment system, previous studies have examined the external validity of LTCI in physical and cognitive disabilities in elderly individuals, and a study determined that the levels of LTCI certification are well associated with the ability to perform activities of daily living assessed by the Barthel Index (Spearman’s coefficient=−0.86) and the Mini-Mental State Examination scores (Spearman’s coefficient =−0.42) (6). Frailty assessed based on the Cardiovascular Health Study criteria (7) was significantly associated with an increased risk of needing LTCI in community-dwelling older adults in Japan.(8) LTCI is obtained by more than 5 million individuals in Japan as they are assessed according to a nationally standardized procedure, including an examination performed by a physician and evaluation of physical and cognitive functions.(9) We believe that LTCI is an effective tool for evaluating functional disability in the elderly. 

Nevertheless, a comparison between LTCI and other scales for patients after HF might provide precise information on post HF-specific functional disability, for example, physical disability caused by dyspnea on exertion. In the present study, we were unable to investigate these possibilities. Therefore, further studies are needed.

We have added these points to the Limitations section (page 18, line 373 to page 19, line 382).

“Fifth, although LTCI is neither a HF specific assessment tool for functional disability, nor validated by some physiological index such as maximum oxygen uptake in exercise, previous studies have examined the external validity of LTCI in physical and cognitive disabilities in elderly individual; a study determined that the levels of LTCI certification are well associated with the ability to perform activities of daily living, assessed by the Barthel Index (Spearman’s coefficient=−0.86) and the Mini-Mental State Examination scores (Spearman’s coefficient =−0.42) (30). Nevertheless, a comparison between LTCI and other scales for patients after HF might provide precise information on post HF-specific functional disability, for example, physical disability caused by dyspnea on exertion. Further studies are therefore needed.”

5. The paragraph on page 5, lines 100-103, is repeated on page 7 lines 128-130.

Our response:

In accordance with your comment, we have deleted the sentence on page 7 lines 128-130.

6. References must include studies that have identified risk factors for the development of disability in patients with heart failure.

Our response:

In accordance with the reviewer’s suggestion, we have added some articles as references (10-12). 

1. Vidán MT, Blaya-Novakova V, Sánchez E, Ortiz J, Serra-Rexach JA, Bueno H. Prevalence and prognostic impact of frailty and its components in non-dependent elderly patients with heart failure. European Journal of Heart Failure. 2016;18(7):869-75.

2. Dunlay SM, Manemann SM, Chamberlain AM, Cheville AL, Jiang R, Weston SA, et al. Activities of daily living and outcomes in heart failure. Circ Heart Fail. 2015;8(2):261-7.

3. Volpato S, Cavalieri M, Sioulis F, Guerra G, Maraldi C, Zuliani G, et al. Predictive value of the Short Physical Performance Battery following hospitalization in older patients. J Gerontol A Biol Sci Med Sci. 2011;66(1):89-96.

 We have added one sentence to the Introduction section (page 5, line 87 to line 88).

 “Studies have identified risk factors for the development of disability in patients with HF. (7, 21, 23)”.

Reviewer: 3

This paper uses a novel dataset to examine the pre-morbid risk factors that contribute to future disability (defined by enrollment in long-term care insurance, or LTCI) after heart failure (HF) in community-dwelling older adults in Japan. The goal is to contribute to earlier detection of high-risk patients and thereby facilitate more timely interventions. From a group of nearly 8,000 patients, the authors identified 96 who developed heart failure over the study period, and split these into two case groups, one of which required LTCI after HF, and the other which did not. Both groups were matched 3-to-one to same age/sex controls who had neither HF nor LTCI during the study period, and cases were compared to controls in both groups. The main results are that hypertension is associated with HF without LTCI, and lower education levels are associated with HF with LTCI.

The topic is an important one given the aging population in Japan, the increasing prevalence of heart failure and the caregiving and quality-of-life costs associated with functional disability. However, the paper does not sufficiently explain the study sample or provide enough information about how to interpret these results with confidence.

Specific Areas for Improvement

Major Issues

1. More discussion of the case-control study design and how this affects how we interpret the results is needed to understand the study’s implications. For example, the case-control study design for the HF-Non LTCI subgroup needs more explanation. The control group has neither HF nor LTCI, while the case group has both; the comparison between them thus speaks to which pre-morbid conditions contribute to the compound outcome of a diagnosis of HF followed by no functional disability. One might suppose that the Table 2 result that hypertension is positively associated with the compound outcome (HF without LTCI) is picking up the association between hypertension and HF. But if this is so, then the same association should be seen in the (HF *with* LTCI) analysis. What does it mean that no such association appears in this second group? How should one interpret these two results together—that hypertension is protective against functional disability among patients with HF? More discussion about how to interpret this and other results in the paper is needed.

Our response:

The purpose of the present study was to examine the association between pre-existing states and deterioration of function after HF. The difference in preclinical risk factors was analyzed among the group for event-free status in the group with no LTCI after HF and the group with LTCI after HF. In recent studies, there have been differences in the risk factors between outcomes for the incidence of HF and prognosis after HF. For example, obesity is a well-known risk factor for HF: obesity increases the risk for incidence of HF (1) but decreases outcomes after HF such as all-cause death or rehospitalization (2, 3). Researchers have recognized the “obesity paradox,” in which it is a risk for HF, but not for prognosis after HF (4). Accumulating evidence accounts for the difference between the two. 

Based on these articles, we need to examine the difference in risk factors between the incidence of HF (the group with no LTCI after HF) and prognosis after HF (the LTCI group after HF). With regard to hypertension, some studies have shown that hypertension increases the risk for HF (13,14). In contrast, one study that examined long-term outcomes after HF showed that hypertension decreased mortality, although all-cause readmission was increased (15). In the present study, we found that the premorbid risk factors differed between groups with LTCI after HF and with no LTCI after HF, i.e., hypertension was related only to no LTCI after HF. Therefore, we could interpret that hyperextension was a risk factor for the incidence of HF, which has already been established as a cardiovascular risk factor. However, because the established cardiovascular risk factor did not have a significant impact on LTCI after HF (OR [95% CI]; hypertension = 1.13 [0.49– 2.57]), we could not determine the protective effect against functional disability among patients with HF. The important risk factors for LTCI after HF were socioeconomic factors instead of conventional cardiovascular risk factors, which might be linked to poor access to medical doctors or poor compliance with sub-optimal treatment.

In accordance with the reviewer’s suggestion, we have added text to the Introduction section (page 5, line 95 to page line 100).

“Recent studies have shown the difference of risk factors between the outcomes for incidence of HF and for prognosis after HF, e.g., “obesity paradox”, where obesity is a risk factor for incident HF, but not for prognosis after HF.(4) The identification of risk factors between no functional disability and functional disability after HF would contribute to earlier detection of high-risk patients and timely interventions.”

In accordance with the reviewer’s suggestion, we have added an explanation in the Discussion section (page 17, line 328 to line 340).

“Recent studies have shown the difference of risk factors between outcomes for incidence of HF and for prognosis after HF.(4) Accumulating evidence accounts for the difference between the two. Some studies have shown that hypertension increases the risk of HF. (13) (14) In contrast, one study that examined long-term outcomes after HF showed that hypertension decreased mortality, although all-cause readmission increased after HF. (15) In the present study, hypertension was related only to no LTCI after HF. We could interpret our results that hyperextension had a high predictive value for the incidence of HF, which has already been described as an established risk factor for HF. In contrast, although we could not determine hypertension as a protective effect against functional disability among patients with HF due to no significant impact on LTCI after HF, the finding that hypertension did not have a significant association with LTCI after HF suggests that social factors, including educational level, have a substantial impact on future functional disability via HF compared with established cardiovascular risk factors.”

2. More detail is needed on the timing of the study and data collection for the reader to follow what has been done. The authors describe the study sample selection process beginning with 7,738 subjects age 65+, and refers to an earlier paper (Ohsawa et al. 2009) for details of the Iwate-KENCO study population. Ohsawa et al. state that the study began in 2002, but it appears that there were many more study participants age 65+ in 2002 (Table 1 in Ohsawa indicates more than 17,000 participants over age 60). To get to the 7,738 starting subjects for the current study, it seems many participants have been excluded from the original study, but there is insufficient detail to understand why that is.

Our response:

The original cohort of the Iwate-KENCO study was recruited from a community-based population living in the Ninohe, Kuji, and Miyako districts of northern Iwate Prefecture, Japan (16). The total number of participants who agreed to join the Iwate-KENCO study in the three districts above was 26,469 (R1 Fig. 1). We used the data of participants in Ninohe and Kuji because the follow-up data on the incidence of congestive heart failure were insufficient in the Miyako district. After excluding participants in Miyako (n=10,542) and those aged 64 years or younger (n=8,189), the number of initial participants was 7,738. 

R1 Figure 1. ABOUT HERE

We have added some comments in the Study population in the Materials and Methods section (page 6, line 110 to line 115).

 “In the original cohort of the Iwate-KENCO study, participants were recruited from a community-based population living in the Ninohe, Kuji, and Miyako districts of Iwate Prefecture, Japan. The total number of participants who agreed to join the Iwate-KENCO study in the three districts above was 26,469. After excluding participants in Miyako (n=10,542) and those aged 64 years or younger (n=8,189), the participants were left 7,738 community-dwelling older adults aged 65 years or older in the Kuji and Ninohe areas (Fig. 1).”

3. The prior point is important because, after all sample selection steps, the current study ends up with fewer than 100 subjects (out of 7,738) who develop heart failure over the 11-year study period; this subgroup is further divided into 52 who did not require long-term care insurance and 44 who did. Some discussion of whether these sample sizes are large enough to detect the kinds of relationships we are looking for is needed. Are the authors confident that the relatively few statistically significant results found are not an artifact of too-small sample sizes?

Our response:

Although the calculation of sample size was important in the observational study, it was not calculated prior to the analysis. In contrast, the sample size were retrospectively calculated, between the two groups with LTCI after HF and without LTCI after HF, respectively (the number of controls per case=3, significance level=0.05, and power=0.8). In the LTCI group after HF, if the same results were obtained (effect size=0.52, lower educational level), a total of 30 cases were needed for the sample size. In the group without LTCI after HF, if the same results were obtained (effect size=0.65, in hypertension), a total of 88 cases were needed as a sample size. Although an estimated total of 118 cases for analyses was slightly smaller than those of cases in the present study (n=96), there was no substantial difference between them. 

A similar analysis was performed to select covariates using forward (Wald) in the Methods section because we considered selecting appropriate covariates. The results were similar in those without LTCI after HF and LTCI after HF (R1 Table 2).

We believe that the significant results in the present study might be robust; in particular, the association with lower educational level in LTCI after HF might be a substantial finding.

R1 Table 2. ABOUT HERE

4. In the Limitations section, the final limitation mentioned is that patients were required to travel to survey venues to participate in the study. This seems especially troubling as it likely limits the degree of disability among the study population, and makes it even more important to describe the study requirements more fully: what exactly was required of patients to participate in the study (frequency of contact, travel requirements such as distance and mode of transport, whether there were supplementary phone or at-home visits etc.)? Without understanding these items, it is difficult to judge how big an impact this limitation might have had.

Our response:

 Subjects who participated in the annual health check-ups of self-employed citizens in the National Healthcare Insurance in Japan from 2002 to 2004 were recruited. Individuals who agreed to participate in our survey took the baseline survey immediately during the terms. These health check-ups were usually conducted in community centers. With regard to LTCI, we followed the reports about LTCI certification, which were assessed and registered by each municipality. We also registered patients with heart failure for hospital inpatients by checking the medical records of the referral hospitals. 

In summary, individuals in the present study participated in the baseline survey just once, and we referred records about LTCI and HF; i.e., the frequency of contact was one time and there were no supplementary phone or at-home visits. Although we could not show the exact distances from participants to the community center, these centers were located in each elementary school district; they are close to each house because these community centers were built for citizens who can access them easily by walking. Therefore, we did not prepare any modes of transportation, such as buses or trains, for participants.

The participants in this study might have had higher health consciousness and better access to health care facilities than those who did not participate in it. This possibility might have led to an underestimation of the present results.

In accordance with the reviewer’s suggestion, we have added explanations about patients to participate in our study (page 6, line 115 to line 119).

“We recruited people who participated in the annual health check-ups of self-employed citizens in the National Healthcare Insurance in Japan from 2002 to 2004. Individuals who agreed to participate in our survey took the baseline survey immediately during the terms. These health check-ups are usually conducted in community centers.”

(page 7, line 142 to line 143)

“We followed the reports about LTCI certification which was assessed and registered by each municipality.”

(page 8, line 166 to line 167)

“We also registered patients with HF for hospital inpatients by checking the medical records of the referral hospitals.”

In addition, we have added some comments in the Limitations section (page 18, line 386 to line 390)

 “Finally, participants in the present study were those who could access the places where our survey was conducted by themselves. These centers were located in each elementary school district; they were close to houses because these community centers were built for citizens who can access them easily by walking.”

Minor Issues

5. The current study also says that subjects were followed for 11 years, but does not describe when the monitoring began and ended, nor how frequently patients were surveyed.

Our response:

 Baseline examinations were performed between 2002 and 2004, and participants completed the baseline survey just once during the terms. 

LTCI is an objectively nationally uniform criterion implemented by the Government of Japan. Certification is determined based on the assessment results of the Certification Committee for Long-term Care Needs in municipalities. The patients’ information about the incidence of cardiovascular diseases was followed, including heart failure registered from hospital records in the cardiovascular disease register program, and LTCI certification information, which was assessed and determined by municipalities for 11 years. The monitoring started from 2002 to 2004 and ended on December 31, 2014. 

We have added text in the Methods section (page 7, line 143 to line 146).

“Certification of LTCI is determined based on assessment results by the Certification Committee for Long-term Care Need in municipalities based on nationally uniform criteria implemented by the Government of Japan.”

(page 8, line 170 to line 171).

“This information was used for both, LTCI certification and the incidence of HF. The monitoring started from 2002 to 2004 and ended on December 31, 2014.”

6. Line 145 suggests that anthropometric measurements were taken repeatedly. If there were multiple regular (say annual) assessments, how were they aggregated over the 11 years of the study?

Our response:

We apologize for this misleading description. Blood pressure was measured twice in the baseline survey. We did not perform repeated or multiple regular assessments in the present cohort study.

In order to avoid misleading, we have revised the sentence in the Measurements section (page 8, line 176 to line 177).

“Blood pressure measurements were performed twice in the baseline survey, and the mean value was used for statistical analyses.”

Miscellaneous Remarks

7. Functional disability was coded as a binary variable—either subjects were enrolled in the LTCI program or not. But apparently the LTCI certification system codes seven different levels of disability. Did the authors consider exploring a “dose effect” – that is, do patients with lower educational levels (or who drink more or smoke more) have higher levels of disability? I wonder if this might sharpen the results.

Our response:

To examine the reviewer’s remarks, the functional disability was divided into three items: those with no LTCI, those with LTCI care need level 1 or less, and those with LTCI care need level 2 or more. LTCI care need level 2 was defined as the requirement of assistance in the group with at least one basic ADL task (17). We performed similar analyses using the LTCI categories in the group with LTCI after HF (R1 Table 3). The number of case groups in LTCI after HR was 40 in the LTCI care need level 1 or less, and 4 in the LTCI care need level 2 or more.

 In Model 3 in the group of LTCI care need level 1 or less, OR was significantly higher for cases with lower educational levels (OR [95% CI]; low educational levels = 4.28 [1.54 - 11.89]). However, we could not find a dose effect pattern according to LTCI categories because we could not calculate ORs in the group of LTCI care need level 2 or more due to an insufficient number of cases. Future studies are needed to identify a dose-response association among the three categories of LTCI.

R1 Table 3. ABOUT HERE

 

References

1. Hubert HB, Feinleib M, McNamara PM, Castelli WP. Obesity as an independent risk factor for cardiovascular disease: a 26-year follow-up of participants in the Framingham Heart Study. Circulation. 1983;67(5):968-77. https://doi.org/10.1161/01.cir.67.5.968 PMID: 6219830

2. Oreopoulos A, Padwal R, Kalantar-Zadeh K, Fonarow GC, Norris CM, McAlister FA. Body mass index and mortality in heart failure: a meta-analysis. Am Heart J. 2008;156(1):13-22. https://doi.org/10.1016/j.ahj.2008.02.014 PMID:18585492

3. Hamaguchi S, Tsuchihashi-Makaya M, Kinugawa S, Goto D, Yokota T, Goto K, et al. Body mass index is an independent predictor of long-term outcomes in patients hospitalized with heart failure in Japan. Circ J. 2010;74(12):2605-11. https://doi.org/10.1253/circj.cj-10-0599 PMID: 21060207

4. Clark AL, Chyu J, Horwich TB. The obesity paradox in men versus women with systolic heart failure. Am J Cardiol. 2012;110(1):77-82. https://doi.org/ 10.1016/j.amjcard.2012.02.050 PMID: 22497678

5. Tsutsui T, Muramatsu N. Care-needs certification in the long-term care insurance system of Japan. J Am Geriatr Soc. 2005;53(3):522-7. http://onlinelibrary.wiley.com/doi/10.1111/j.1532-5415.2005.53175.x/abstract PMID: 15743300

6. Arai Y, Zarit SH, Kumamoto K, Takeda A. Are there inequities in the assessment of dementia under Japan's LTC insurance system? Int J Geriatr Psychiatry. 2003;18(4):346-52. https://onlinelibrary.wiley.com/doi/abs/10.1002/gps.836 PMID: 12673612

7. Fried LP, Tangen CM, Walston J, Newman AB, Hirsch C, Gottdiener J, et al. Frailty in older adults: evidence for a phenotype. J Gerontol A Biol Sci Med Sci. 2001;56(3):M146-56. http://biomedgerontology.oxfordjournals.org/content/56/3/M146.full.pdf PMID: 11253156

8. Chen S, Honda T, Narazaki K, Chen T, Kishimoto H, Kumagai S. Physical Frailty and Risk of Needing Long-Term Care in Community-Dwelling Older Adults: a 6-Year Prospective Study in Japan. J Nutr Health Aging. 2019;23(9):856-861. http://dx.doi.org/10.1007/s12603-019-1242-6

9. Ashida T, Kondo N, Kondo K. Social participation and the onset of functional disability by socioeconomic status and activity type: The JAGES cohort study. Prev Med. 2016;89:121-8. https://www.sciencedirect.com/science/article/pii/S0091743516300834?via%3Dihub PMID: 27235600

10. Vidán MT, Blaya-Novakova V, Sánchez E, Ortiz J, Serra-Rexach JA, Bueno H. Prevalence and prognostic impact of frailty and its components in non-dependent elderly patients with heart failure. European Journal of Heart Failure. 2016;18(7):869-875. https://onlinelibrary.wiley.com/doi/abs/10.1002/ejhf.518

11. Dunlay SM, Manemann SM, Chamberlain AM, Cheville AL, Jiang R, Weston SA, et al. Activities of daily living and outcomes in heart failure. Circ Heart Fail. 2015;8(2):261-7. https://www.ncbi.nlm.nih.gov/pmc/articles/PMC4366326/pdf/nihms653469.pdf PMID: 25717059

12. Volpato S, Cavalieri M, Sioulis F, Guerra G, Maraldi C, Zuliani G, et al. Predictive value of the Short Physical Performance Battery following hospitalization in older patients. J Gerontol A Biol Sci Med Sci. 2011;66(1):89-96. https://doi.org/10.1093/gerona/glq167 PMID: 20861145

13. Levy D, Larson MG, Vasan RS, Kannel WB, Ho KK. The progression from hypertension to congestive heart failure. Jama. 1996;275(20):1557-62. PMID: 8622246

14. McKee PA, Castelli WP, McNamara PM, Kannel WB. The natural history of congestive heart failure: the Framingham study. N Engl J Med. 1971;285(26):1441-6. https://www.nejm.org/doi/full/10.1056/NEJM197112232852601?url_ver=Z39.88-2003&rfr_id=ori%3Arid%3Acrossref.org&rfr_dat=cr_pub%3Dpubmed PMID: 5122894

15. Curtis LH, Greiner MA, Hammill BG, Kramer JM, Whellan DJ, Schulman KA, et al. Early and Long-term Outcomes of Heart Failure in Elderly Persons, 2001-2005. Archives of Internal Medicine. 2008;168(22):2481-2488. https://doi.org/10.1001/archinte.168.22.2481

16. Takahashi S, Tanaka F, Yonekura Y, Tanno K, Ohsawa M, Sakata K, et al. The urine albumin-creatinine ratio is a predictor for incident long-term care in a general population. PLoS One. 2018;13(3):e0195013. https://www.ncbi.nlm.nih.gov/pmc/articles/PMC5874057/pdf/pone.0195013.pdf PMID: 29590199

17. Bando S, Tomata Y, Aida J, Sugiyama K, Sugawara Y, Tsuji I. Impact of oral self-care on incident functional disability in elderly Japanese: the Ohsaki Cohort 2006 study. BMJ open. 2017;7(9):e017946. http://search.ebscohost.com/login.aspx?direct=true&db=cmedm&AN=28928197&lang=ja&site=ehost-live PMID: 28928197

---

## [Decision Letter · Decision Letter 1]

27 May 2021

Low educational level increases functional disability risk subsequent to heart failure in Japan: on behalf of the Iwate KENCO study group

PONE-D-21-00523R1

Dear Dr. Takahashi,

We’re pleased to inform you that your manuscript has been judged scientifically suitable for publication and will be formally accepted for publication once it meets all outstanding technical requirements.

Kind regards,

Antonio Palazón-Bru, PhD

Academic Editor

PLOS ONE

Additional Editor Comments (optional):

Reviewers' comments:

Reviewer's Responses to Questions

**Comments to the Author**

1. If the authors have adequately addressed your comments raised in a previous round of review and you feel that this manuscript is now acceptable for publication, you may indicate that here to bypass the “Comments to the Author” section, enter your conflict of interest statement in the “Confidential to Editor” section, and submit your "Accept" recommendation.

Reviewer #1: All comments have been addressed

Reviewer #2: All comments have been addressed

Reviewer #3: All comments have been addressed

2. Is the manuscript technically sound, and do the data support the conclusions?

Reviewer #1: Yes

Reviewer #2: Yes

Reviewer #3: Yes

3. Has the statistical analysis been performed appropriately and rigorously? 

Reviewer #1: Yes

Reviewer #2: Yes

Reviewer #3: Yes

4. Have the authors made all data underlying the findings in their manuscript fully available?

Reviewer #1: No

Reviewer #2: Yes

Reviewer #3: Yes

5. Is the manuscript presented in an intelligible fashion and written in standard English?

Reviewer #1: Yes

Reviewer #2: Yes

Reviewer #3: Yes

6. Review Comments to the Author

Reviewer #1: (No Response)

Reviewer #2: (No Response)

Reviewer #3: (No Response)

7. PLOS authors have the option to publish the peer review history of their article (what does this mean?). If published, this will include your full peer review and any attached files.

Reviewer #1: No

Reviewer #2: No

Reviewer #3: No

---

## [Editor Report · Acceptance letter]

31 May 2021

PONE-D-21-00523R1 

Low educational level increases functional disability risk subsequent to heart failure in Japan: on behalf of the Iwate KENCO study group 

Dear Dr. Takahashi:

I'm pleased to inform you that your manuscript has been deemed suitable for publication in PLOS ONE. Congratulations! Your manuscript is now with our production department. 

Kind regards, 

on behalf of

Dr. Antonio Palazón-Bru 

Academic Editor

PLOS ONE